# BAYESIAN METRIC LEARNING FOR ROBUST TRAINING OF DEEP MODELS UNDER NOISY LABELS

## ABSTRACT

Label noise is a natural event of data collection and annotation and has been shown to have significant impact on the performance of deep learning models regarding accuracy reduction and sample complexity increase. This paper aims to develop a novel theoretically sound Bayesian deep metric learning that is robust against noisy labels. Our proposed approach is inspired by a linear Bayesian large margin nearest neighbor classification, and is a combination of Bayesian learning, triplet loss-based deep metric learning and variational inference frameworks. We theoretically show the robustness under label noise of our proposed method. The experimental results on benchmark data sets that contain both synthetic and realistic label noise show a considerable improvement in the classification accuracy of our method compared to the linear Bayesian metric learning and the point estimate deep metric learning.

## 1 INTRODUCTION

Deep learning has been shown as a dominant learning framework in various domains of machine learning and computer vision. One of the major limitations of deep learning is that it often requires relatively clean data sets that do not contain label noise naturally caused by human labeling errors, measurement errors, subjective biases and other issues (Frénay et al., 2014; Ghosh et al., 2017; Algan & Ulusoy, 2019). The performance of a machine learning method can be significantly affected by noisy labels both in terms of the reduction in the accuracy rate and the increase in sample complexity. Particularly for deep learning, a deep neural network (DNN) can generalize poorly when trained with noisy training sets which contain high proportion of noisy labels since a DNN can over-fit those noisy training data sets (Zhang et al., 2016; Algan & Ulusoy, 2020). Developing deep learning methods that can perform well on noisy training data is essential since it can enable the use of deep models in many real-life applications.

There have been several approaches proposed to handle learning issues caused by label noise, for example: data cleaning (Angelova et al., 2005; Chu et al., 2016), label correction (Reed et al., 2014), additional linear correction layers (Sukhbaatar et al., 2014), dimensionality-driven learning (Ma et al., 2018), bootstrapping (Reed et al., 2014), curriculum learning-model based approach such as MentorNet (Jiang et al., 2018) or CoTeaching (Han et al., 2018), loss correction (or noise-tolerant loss) (Masnadi-Shirazi & Vasconcelos, 2009; Ghosh et al., 2017; Zhang & Sabuncu, 2018; Thulasidasan et al., 2019; Ma et al., 2020), or a combination of the techniques above (Li et al., 2020; Nguyen et al., 2019). Relevant to this paper is an existing theoretically sound approach: Bayesian large margin nearest neighbor classification (BLMNN) (Wang & Tan, 2018) that employs Bayesian inference to improve the robustness of a point estimation-based linear metric learning method. BLMNN then introduces a method to approximate the posterior distribution of the underlying distance parameter given the triplet data by using the stochastic variational inference. More importantly, BLMNN (Wang & Tan, 2018) also provides a theoretical guarantee about the robustness of the method, which says that it can work with non-uniform label noise. Although BLMNN has been mathematically shown to be robust against label noise, it only focuses on a simple linear Mahalanobis distance that can not capture the nonlinear relationships of data points in deep metric learning (Lu et al., 2017).

In this paper, we introduce a Bayesian deep metric learning framework that is robust against noisy labels. Our proposed method (depicted in Fig. 1) is inspired by the BLMNN (Wang & Tan, 2018),

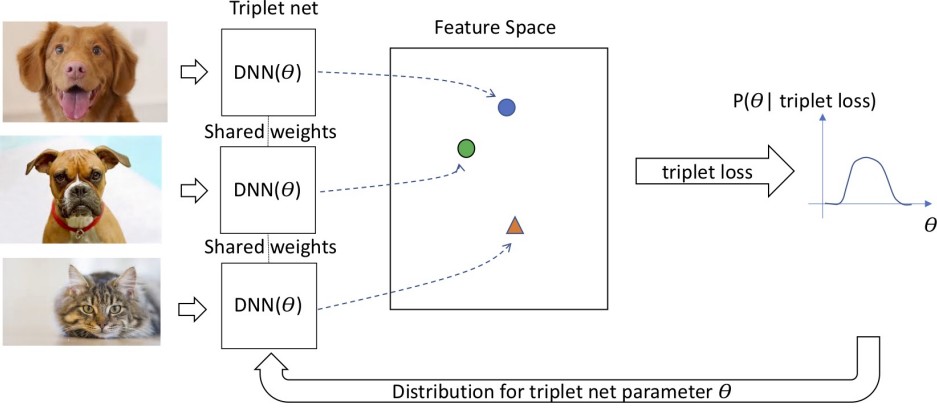

Figure 1: An overview of our proposed Bayesian Deep Metric Learning method

deep metric learning (Hoffer & Ailon, 2015; Hu et al., 2015; Wang et al., 2017; Lu et al., 2017; Do et al., 2019), and Bayes by Backprop (Blundell et al., 2015). Compared to the BLMNN that only considers a linear metric learning, our framework can handle non-linear deep metric learning, which is useful for many real-life applications. Moreover, directly applying the variational Bayes learning (Wang & Tan, 2018) in deep learning is challenging since it requires sampling from a distribution of the neural network parameters. Instead, we adapt the variational inference by Blundell et al. (Blundell et al., 2015), which allows to efficiently sample the parameters of a Bayes neural networks by using a backpropagation-compatible algorithm. We also theoretically show the robustness of our proposed method when working with label noise. The experimental results on several noisy data sets show that our novel proposed method can generalize better compared to the linear BLMNN (Wang & Tan, 2018) and the point estimation-based deep metric learning (Hoffer & Ailon, 2015; Lu et al., 2017), especially when the noise level increases.

It is important to emphasize that the motivation of our method is to produce a better calibrated model that is more robust to noisy label training, and, as a result, less likely to overfit the training set than the linear BLMNN (Wang & Tan, 2018) and the point estimation-based deep metric learning (Hoffer & Ailon, 2015; Lu et al., 2017). Therefore this is a paper that introduces a new theoretical framework to solve noisy label learning instead of presenting method that is competitive against the best approaches of the field (such as Mentornet (Jiang et al., 2018) or Co-teaching (Han et al., 2018)) in large-scale datasets (e.g., webvision (Li et al., 2017) and Clothing 1M (Xiao et al., 2015)). Furthermore, deep metric learning has been in fact considered in the Bayesian settings before (Ishfaq et al., 2018; Karaletsos et al., 2015), and recently, in (Lin et al., 2018), but not in the context of noisy labels. Consequently, our proposed framework can be used by other methods that can deal with noisy label learning, but the extension of those methods using our proposed approach is out of the scope of this paper.

## 2 RELATED WORK

### 2.1 POINT ESTIMATION-BASED DISTANCE METRIC LEARNING

The goal of distance metric learning (or metric learning) is to learn a distance function to measure the similarity between training samples. Metric learning has been shown to have great success in many visual applications such as face recognition, image classification, visual search, visual tracking, and person re-identification (Lu et al., 2017). In principle, a supervised metric learning method aims to learn a distance metric which pulls together samples from the same class while pushing away those from different classes. Based on the complexity of the distance, metric learning can be classified into two types: linear, focusing on linear distance (e.g., Mahalanobis), which often suffers from the nonlinear relationship of data points (Lu et al., 2017); and non-linear, which nowadays is mostly based on deep learning.

Deep metric learning (DML) is motivated by the fact that deep learning is an effective solution to a non-linear transformation of input samples (Lu et al., 2017; Do et al., 2019). The key idea of DML is to explicitly learn a set of hierarchical non-linear transformations to map input data points into a feature space that is used for comparing or matching these data points in a more effective manner. DML unifies feature learning and metric learning into a joint learning framework. DML is shown to be more advantageous compared to traditional models, for example, with respect to classification performance since DML does not rely on the classification layer of a trained model that often depends on the type of problems (Lu et al., 2017; Do et al., 2019). Relevant to this paper is the triplet loss-based metric learning (Lu et al., 2017; Schroff et al., 2015), in which the original training data set is represented by a set of independent triplets, formed by an anchor sample, one sample of the same class and another sample from a different class. The training process is then performed by minimizing the triplet loss to simultaneously approximate the anchor to the sample with the same class and separate the anchor from the sample of different class.

## 2.2 ROBUST LINEAR DISTANCE METRIC LEARNING VIA BAYESIAN INFERENCE

One drawback of the point estimation DML is that it is likely to over-fit the noisy labels (Wang & Tan, 2014; 2018). Wang et al. (2016) introduces the Deep Stochastic Neighbor Compression (DSNC) method that aims to jointly learn a nonlinear transformation that preserves the neighborhood of the data, and a compressed version of the training data set (Ahmed et al.). DSNC is also robust against label noise. Motivated by the fact that Bayesian learning is a good choice for robust learning (Zhu et al., 2014; Yang et al., 2012), Wang & Tan (2018) introduced the theoretically sound Bayesian large margin nearest neighbor classification (BLMNN) to improve the robustness of the linear metric learning under the presence of label noise. The BLMNN framework represents the large margin nearest neighbor classification using a linear metric learning in the form of a variational Bayes method that takes the prior distribution of the transformation matrix into account and estimate the posterior distribution via stochastic variational inference (SVI). BLMNN provides mathematical definitions of the noisy label triplet (a type of non-uniform label noise) and the $\beta$-robust algorithm against noisy label triplets. More importantly, a theoretically guarantee of the robustness of BLMNN is also provided in (Wang & Tan, 2018). Although BLMNN efficiently addresses the training issue caused by the noisy label, one key limitation of BLMNN is that it is based on a linear metric transformation that cannot capture the nonlinear relationships of data points in deep metric learning. However, the extension of BLMNN to the non-linear case is not straightforward due to the complexity of the posterior distribution estimation as well as the sampling process of high-dimensional parameters, and such extension is the main target of our paper.

## 2.3 BAYES BY BACKPROP

Bayesian neural networks (MacKay, 1995; Neal, 2012; Gal & Ghahramani, 2015) aim to estimate the posterior distribution of the network parameters given the training data. However, that inference framework is often intractable, especially when working with high dimensional parameters (Blundell et al., 2015). Moreover, exactly calculating the posterior of the weights is challenging since it requires the integration that is known to be computationally expensive. Blundell et al. (Blundell et al., 2015) introduced the "backpropagation-compatible" Bayes by Backprop method for estimating the posterior distribution of the network parameter. That method is inspired by the variational free energy inference in which the exact posterior is approximated by a variational distribution by solving an optimization problem. In principle, Bayes by Backprop can directly work on the network parameter by minimizing a compression cost function (Blundell et al., 2015).

Our proposed method is inspired by (Wang & Tan, 2018), where we replace the linear Mahalanobis metric by a non-linear deep metric. Our novel proposed method therefore leverages the good performance of the deep metric learning (Lu et al., 2017) and the robustness to label noise of a Bayesian framework (Yang et al., 2012; Wang & Tan, 2018). We first represent a triplet-based deep metric learning using Bayesian inference. By replacing the softmax loss by a triplet loss (and so learning a metric), we impose more strict constraints, where points of the same class are forced to collapse into a small region of the feature space instead of just belonging to regions within the class boundaries. We argue that this more strict constraints has the potential to introduce more robust feature spaces for classification under noisy label training. To approximate the posterior of the network parameters, we then employ the efficient variational framework (Blundell et al., 2015) that allows to sample

from a Bayesian neural network using the backpropagation framework. We theoretically show that our proposed method is robust against noisy triplet relying on a Bayesian inference framework.

## 3 METHODOLOGY

In this section, we explain our proposed method. Then, we theoretically show the robustness of that method under noisy label triplet.

### 3.1 DEEP METRIC LEARNING WITH TRIPLET LOSS

Let us denote the original data set by $\mathcal{T} = \{(\mathbf{x}_i, y_i)\}_{i=1}^N$, where $\mathbf{x}_i \in \mathbb{R}^D$ is a sample and $y_i \in \mathcal{Y} \subset \mathbb{N}$ is its corresponding label. Let $\mathcal{S} = \{(i,j)|(\mathbf{x}_i, y_i), (\mathbf{x}_j, y_j) \in \mathcal{T}$ and $y_i = y_j\}$ and $\mathcal{D} = \{(i,j)|(\mathbf{x}_i, y_i), (\mathbf{x}_j, y_j) \in \mathcal{T}$ and $y_i \neq y_j\}$ denote the set of pairs of the data points with the same labels and the set of pairs with different labels, then a triplet is defined by $z = (i,j,l)$, where $(i,j) \in \mathcal{S}$, and $(i,l) \in \mathcal{D}$. A triplet-based deep metric learning framework (Hoffer & Ailon, 2015; Lu et al., 2017; Do et al., 2019) aims to minimize the following triplet loss function for the triplet $z = (i,j,l)$,

$$\mathcal{L}_{tri}^{(i,j,l)}(\theta) = h(\tau + d_f^2(\mathbf{x}_i, \mathbf{x}_j) - d_f^2(\mathbf{x}_i, \mathbf{x}_l)), \tag{1}$$

where the deep metric is defined by

$$d_f^2(\mathbf{x}_i, \mathbf{x}_j) = \|f(\mathbf{x}_i, \theta) - f(\mathbf{x}_j, \theta)\|_2^2, \tag{2}$$

with $f(\cdot, \theta)$ denoting the network function parameterized by $\theta$, and $h(a) = \max(0, a)$, $a \in \mathbb{R}$ representing the hinge loss function (with $\tau > 0$ being a margin between $d_f(\mathbf{x}_i, \mathbf{x}_j)$ and $d_f(\mathbf{x}_i, \mathbf{x}_l)$).

### 3.2 BAYESIAN DEEP METRIC LEARNING

Let us denote the training data set by $\mathcal{Z} = \{z_k\}_{k=1,\ldots,|\mathcal{Z}|}$, which contains independent triplets $z = (i,j,l)$ and the network parameter by $\theta \in \mathbb{R}^P$. According to the large margin principle (Wang & Tan, 2018), the likelihood function $p(\mathcal{Z}|\theta)$ can be defined as follows:

$$\begin{aligned} p(\mathcal{Z}|\theta) &= \prod_{(i,j,l)\in\mathcal{Z}} p(\mathbf{x}_i, \mathbf{x}_j, \mathbf{x}_l, y_i, y_j, y_l|\theta) \\ &= C \prod_{(i,j,l)\in\mathcal{Z}} \exp(-2 \cdot \max(1 + d_f^2(\mathbf{x}_i, \mathbf{x}_j) - d_f^2(\mathbf{x}_i, \mathbf{x}_l), 0)), \end{aligned} \tag{3}$$

where $C$ is a normalising constant. Without loss of generality, we assume that the prior of the network parameter is a Gaussian with mean $\mu_0$ and covariance matrix $\mathbf{V}_0$, i.e, $p(\theta) = \mathcal{N}(\theta|\mu_0, \mathbf{V}_0)$. By using the Bayes' rule, the posterior distribution of the parameter $\theta$ can be represented by

$$p(\theta|\mathcal{Z}) \propto p(\mathcal{Z}|\theta) \times p(\theta). \tag{4}$$

Naively estimating the posterior in equation 4 is often intractable, especially for the high dimensional parameter space of deep models. Hence, we employ the variational learning approach (Blundell et al., 2015; Blei et al., 2017) to approximate the posterior distribution in equation 4. In particular, the goal of that variational method is to estimate the variational parameter $\lambda$ of a distribution for the network parameters, denoted by $q(\theta|\lambda)$ that minimizes the Kullback-Leiber (KL) divergence between the variational distribution and the true posterior, i.e.,

$$\begin{aligned} \lambda^* &= \arg\min_\lambda \text{KL}(q(\theta|\lambda)\|p(\theta|\mathcal{Z})) \\ &= \arg\min_\lambda \text{KL}(q(\theta|\lambda)\|p(\theta)) - \mathbb{E}_{q(\theta|\lambda)}\{\log p(\mathcal{Z}|\theta)\}. \end{aligned} \tag{5}$$

The objective function in equation 5 embodies a trade-off between the complexity of the data and the simplicity of the prior. Blundell et al. (Blundell et al., 2015) pointed out that directly evaluating that cost function is computationally expensive (Blundell et al., 2015), so they introduced an unbiased Monte-Carlo approximation of the exact cost function in equation 5 with the following function:

$$\mathcal{F}(\mathcal{Z}, \theta) \approx \sum_{i=1}^n \log q(\theta^{(i)}|\lambda) - \log(p(\theta^{(i)})) - \log p(\mathcal{Z}|\theta^{(i)}), \tag{6}$$

---

**Algorithm 1** Bayesian Deep Metric Learning (BDML)

---
    Establish the triplet training set $\mathcal{Z}$
    Initialize the variational parameter $\lambda = (\mu, \rho)$
    **repeat**
        Sample $\varepsilon \sim \mathcal{N}(0, I)$
        Compute $\theta$ with equation 7
        Compute the likelihood $p(\mathcal{Z}|\theta)$ with equation 3
        Compute function $h(\theta, \lambda)$ with equation 9
        Update $\mu, \rho$ with equation 8
    **until** convergence

---

where $\theta^{(i)} \sim q(\theta|\lambda)$ is the $i$-th Monte Carlo sample of the variational posterior distribution assumed to be a diagonal Gaussian distribution, i.e., $\theta \sim \mathcal{N}(\mu, \sigma^2 \mathbf{I})$, where $\mathbf{I}$ is the identity matrix, then a sample of the weights $\theta$ can be obtained by

$$\theta \sim \mu + \log(1 + \exp(\rho)) \circ \varepsilon, \tag{7}$$

where $\sigma = \log(1 + \exp \rho)$, the variational posterior parameters are $\lambda = (\mu, \rho)$, with $\mu \in \mathbb{R}^P$ and $\rho \in \mathbb{R}$, $\circ$ denoting point-wise multiplication, and $\varepsilon \sim \mathcal{N}(0, \mathbf{I})$. The variational parameters $\lambda = (\mu, \rho)$ are updated by (Blundell et al., 2015):

$$
\begin{aligned}
\mu &\leftarrow \mu - \alpha \nabla_\mu h(\theta, \lambda) \\
\rho &\leftarrow \rho - \alpha \nabla_\rho h(\theta, \lambda),
\end{aligned}
\tag{8}
$$

where $\alpha$ is the step size, and

$$h(\theta, \lambda) = \log q(\theta|\lambda) - \log p(\theta)p(\mathcal{Z}|\theta). \tag{9}$$

The gradients in equation 8 can be estimated as follows (Blundell et al., 2015):

$$
\nabla_\mu h(\theta, \lambda) = \frac{\partial h(\theta, \lambda)}{\partial \theta} + \frac{\partial h(\theta, \lambda)}{\partial \mu}, \tag{10}
$$

$$
\nabla_\rho h(\theta, \lambda) = \frac{\partial h(\theta, \lambda)}{\partial \theta} \frac{\varepsilon}{1 + \exp(-\rho)} + \frac{\partial h(\theta, \lambda)}{\partial \rho}. \tag{11}
$$

The whole training process of our proposed method is presented in Algorithm 1.

### 3.3 ROBUSTNESS TO LABEL NOISE

This subsection first introduces the definition of a noisy label triplet and the robustness of a learning algorithm under that label noise. We then theoretically show the robustness of our proposed method against noisy labels.

**Definition 1.** A triplet $z = (i, j, l)$ of data points is defined as a label noisy triplet (Wang & Tan, 2018) if: 1) $\mathbf{x}_j, \mathbf{x}_l \in \mathcal{N}_i$–the set of neighbors (Weinberger & Saul, 2009) of $\mathbf{x}_i$, where $\mathcal{N}_i = \{\mathbf{x} \in \mathbb{R}^D : \|\mathbf{x} - \mathbf{x}_i\| \leq \delta\}$, with $\delta > 0$ being sufficiently small; 2) $y_{ij}(1 - y_{il}) = 1$, where $y_{ij} = \begin{cases} 1, & \text{if } y_i = y_j \\ 0, & \text{otherwwise} \end{cases}$; and
3) $d_f^2(\mathbf{x}_i, \mathbf{x}_j) - d_f^2(\mathbf{x}_i, \mathbf{x}_l) \geq C_d$, with $C_d > 0$ denoting a threshold.

**Definition 2.** Let $\mathcal{Z}$ be an arbitrary training set and $z'$ a label noisy triplet. Then a learning algorithm $\mathcal{A}$ is *$\beta$-robust against label noise in the Bayesian inference sense* if

$$|\log p(\theta|\mathcal{Z}, z') - \log p(\theta|\mathcal{Z})| \leq \beta. \tag{12}$$

**Lemma 1.** *Let $\mathcal{Z}$ be an arbitrary training set (in the sense that it may or may not contain any noisy label triplet). Assuming that $z' = (i, j, l)$ is a noisy label triplet with the corresponding threshold $C_d$ in Definition 1, and the normalising constant of the likelihood $p(\mathcal{Z}|\theta)$, $C$ is defined in equation 3 such that $C \leq \exp(2(C_d + 1))$. Then*

$$|\log p(\mathcal{Z}, z'|\theta) - \log p(\mathcal{Z}|\theta)| \leq 2(C_d + 1) - \log C. \tag{13}$$

*Proof.* Suppose that $p(\theta) \neq 0$. Given the i.i.d. assumption from equation 3 we have

$$p(\mathcal{Z}, z'|\theta) = p(\mathcal{Z}|, \theta) \times p(z'|\theta). \tag{14}$$

Thus

$$\log p(\mathcal{Z}, z'|\theta) = \log p(\mathcal{Z}|\theta) + \log p(z'|\theta). \tag{15}$$

Note that $0 < p(z'|\theta) \leq 1$, we have $\log p(z'|\theta) \leq 0$, and then

$$|\log p(\mathcal{Z}, z'|\theta) - \log p(\mathcal{Z}|\theta)| = -\log p(z'|\theta). \tag{16}$$

Since $z' = (i, j, l)$ is a noisy label triplet (Definition 1), we obtain

$$1 + d_f^2(\mathbf{x}_i, \mathbf{x}_j) - d_f^2(\mathbf{x}_i, \mathbf{x}_l) \geq 1 + C_d > 0, \tag{17}$$

then, by combining with the corresponding likelihood in equation 3, we obtain

$$-\log p(z'|\theta) \leq -\log C + \log(\exp(2(1 + C_d))) \leq 2(C_d + 1) - \log C. \tag{18}$$

The resutl in equation 13 is obtained by integrating equation 18 into equation 16. $\qquad \square$

**Remark 1.** Assuming that the prior $p(\theta)$ is arbitrary, but fixed and that the noisy triplet $z'$ and the original data $\mathcal{Z}$ are sampled from the same distribution, by using Bayes' rule we then get a similar estimate for the posterior distribution, i.e.,

$$|\log p(\theta|\mathcal{Z}, z') - \log p(\theta|\mathcal{Z})| \leq 2(C_d + 1) - \log C = \beta. \tag{19}$$

Hence, according to the Definition 2 and equation 19, Algorithm 1 is Bayesian robust against noisy label triplet.

**Theorem 1.** Given Lemma 1, and supposing that we have $N$ noisy label triplets $z'_1, \ldots, z'_N$ that are conditionally independent given the parameter $\theta$, then

$$|\log p(\theta|\{\mathcal{Z}, z'_1, \ldots z'_N\}) - \log p(\theta|\mathcal{Z})| \leq N\beta. \tag{20}$$

*Proof.* By applying the result in Equation 19 of Lemma 1 we get

$$|\log p(\theta|\{\mathcal{Z}, z'_1\}) - \log p(\theta|\mathcal{Z})| \leq \beta;$$
$$|\log p(\theta|\{\mathcal{Z}, z'_1, z'_2\}) - \log p(\theta|\mathcal{Z}, z'_1)| \leq \beta;$$
$$\cdots$$
$$|\log p(\theta|\{\mathcal{Z}, z'_1, \ldots z'_N\}) - \log p(\theta|\{\mathcal{Z}, z'_1, \ldots z'_{N-1}\})| \leq \beta. \tag{21}$$

Moreover, by using the triangle inequality we obtain equation 20. $\qquad \square$

Given an arbitrary training set, the theoretical result in Lemma 1 indicates that our proposed method is $\beta$-robust against a noisy label triplet $z'$. However, in many real-life applications, the noise level is often much larger, that is, we have to deal with a number of noisy triplets. The theoretical result in Theorem 1, which is an extension of equation 19, shows the robustness of our proposed novel method given a particular noise level represented by $N$ noisy triplets. Moreover, it would be interesting to clarify that although the definition of the robustness of a metric learning algorithm had been dicussed, for example, in (Bellet & Habrard, 2015; Huai et al., 2019), these papers do not take noisy labels into account.

## 4 EXPERIMENTS AND RESULTS

In this section, we quantitatively evaluate our proposed method based on the experiments conducted on several benchmark data sets that contain different types of label noise. In particular, our proposed method Bayesian deep metric learning (BDML) is compared with BLMNN and the deterministic triplet-based deep metric learning (DML) with respect to classification performance on the data sets MNIST (LeCun et al., 1998) and CIFAR-10 (Krizhevsky et al., 2012) with synthetic label noise, and face retrieval results on MS-Celeb-1M (Guo et al., 2016) with realistic noisy labels. In all our experiments, we adopt a simple network architecture for deep learning models with two fully-connected hidden layers of $512$-dimension and $Tanh$ activation function between intermediate layers. Our

Table 1: Classification results for MNIST (mean±stdev) after five runs for different rates of uniform label noise. The best result per column is in bold

| Noise Rate | 0% | 10% | 20% | 30% | 50% |
|---|---|---|---|---|---|
| PCA | $97.87 \pm 0.09$ | $95.35 \pm 0.30$ | $89.56 \pm 0.53$ | $80.70 \pm 0.47$ | $59.02 \pm 0.65$ |
| BLMNN | $\mathbf{98.98 \pm 0.08}$ | $97.08 \pm 0.20$ | $92.25 \pm 0.51$ | $84.47 \pm 0.47$ | $63.60 \pm 0.81$ |
| DML | $98.66 \pm 0.07$ | $\mathbf{97.76 \pm 0.20}$ | $\mathbf{95.34 \pm 0.35}$ | $91.40 \pm 0.52$ | $71.86 \pm 0.47$ |
| BDML (ours) | $98.79 \pm 0.09$ | $97.60 \pm 0.18$ | $95.25 \pm 0.34$ | $\mathbf{91.59 \pm 0.36}$ | $\mathbf{73.60 \pm 0.6}$ |

proposed framework consists of the following steps: (i) extract features by an unsupervised method; (ii) apply PCA to project data points onto a smaller dimensional space; (iii) standardize features to standard Normal distribution, then use these features for comparing methods.

The values of these hyper-parameters were selected by using $10\%$ of training data as a hold-out set for validation. Performance is calculated by 3-NN classifier, each experiment is repeated five times, and the comparison results with respect to the top-1 classification accuracy (mean±stdev) as a function of the percentage of noisy labels in the training set are reported.

## 4.1 IMAGE CLASSIFICATION WITH SYNTHETIC LABEL NOISE

### 4.1.1 SYMMETRIC NOISE

**The MNIST data set (LeCun et al., 1998)** contains $28 \times 28$ black and white handwritten digits in 10 classes, (with 50000 training and 10000 testing samples). The new training data set of MNIST is generated by randomly stratified sub-sampling 5000 training data points from the original training data (i.e., 500 samples for each class), and the standard test set is unchanged – this enables a fair comparison between different methods. The 300-dimensional feature vector for each data point is extracted by using the unsupervised representation learning model CSVDDNet (Wang & Tan, 2018; 2016). Uniform (or symmetric) label noise with corresponding level in the set $\{0, 10\%, 20\%, 30\%\}$ is then injected into the new training data set by flipping the labels by the respective proportion of data points.

*Hyper-parameters settings:* for the *BDML model* we run 50 epochs, set the learning rate to 0.0001 with decay rate 0.5 at epochs 30, prior distribution for weights and bias are Gaussian distributions with mean 0 and standard deviation 1 for weights, 2 for bias; batch size is set to 1024; for the *DML model*, we run 50 epochs, learning rate is set to 0.00001 with decay rate 0.5 at epoch 30, batch size is set to 128. For the *BLMNN model*, we follow the hyper-parameter setting suggested by authors (Wang & Tan, 2018)

Table 1 shows that in the case of clean data set, BLMNN, which employs a linear transformation, produces slightly better classification performance than both DML and BDML. However, when the noise level increases, deep metric-based methods outperform linear model. In particular, in the case of $30\%$ noise, our method outperforms BLMNN with a large margin ($7.12\%$), and in the most extreme case with $50\%$ of noise, BDML produces better results than both BLMNN and DML.

**The CIFAR-10 data set (Krizhevsky et al., 2012)** consists of $32 \times 32$ colour images (with 50000 training and 10000 testing samples, and 10 classes). The new training data set is generated by randomly sub-sampling 5000 data points from original training data (i.e., 500 samples for each class), while the standard test set remained. The 2048-dimensional feature of each data point is extracted with the recently proposed Simple Framework for Contrastive Learning of Visual Representations (SimCLR) (Chen et al., 2020) for the feature extraction process.

*Hyper-parameters settings:* for the *BDML model*, we run 50 epochs, set learning rate to 0.0001 with decay rate 0.5 at epoch 30, prior distribution for weights and bias are Gaussian distributions with mean 0 and standard deviation 1 for weights, 2 for bias. For *DML model*, we set learning rate to 0.00001 with decay rate 0.5 at epoch 30 and 50, batch size is set to 512. For *BLMNN model*, we use the prior mean 0.00001 and variance 0.001.

We report the performance of three models with respect to different noise levels in the set $\{0, 10\%, 20\%, 30\%, 50\%\}$ in Table 2. It is clear that the use of a deep metric learning gives better classification results comparing with linear method, especially when the noise level increases. To be more specific, the gap increase from $2.67\%$ at clean label to $13.66\%$ at $50\%$ label noise. More importantly, the results in that table consistently show that our proposed BDML method, which adopt Bayesian

Table 2: Classification results for CIFAR-10 (mean±stdev) after five runs for different rates of uniform label noise. The best result per column is in bold

| Noise Rate | 0% | 10% | 20% | 30% | 50% |
|---|---|---|---|---|---|
| PCA | $76.64 \pm 0.41$ | $72.98 \pm 0.29$ | $67.83 \pm 0.54$ | $60.91 \pm 0.18$ | $44.06 \pm 0.73$ |
| BLMNN | $79.96 \pm 0.17$ | $76.56 \pm 0.28$ | $71.18 \pm 0.53$ | $64.89 \pm 0.38$ | $47.74 \pm 0.15$ |
| DML | $81.81 \pm 0.30$ | $80.21 \pm 0.29$ | $77.90 \pm 0.34$ | $74.05 \pm 0.24$ | $59.76 \pm 0.64$ |
| BDML (ours) | $\mathbf{82.63 \pm 0.12}$ | $\mathbf{81.01 \pm 0.27}$ | $\mathbf{78.81 \pm 0.25}$ | $\mathbf{75.17 \pm 0.34}$ | $\mathbf{61.40 \pm 0.46}$ |

Table 3: Classification results for CIFAR-10 (mean±stdev) after five runs for different class-conditional label noises, noise rates are presented as a vector with length equals to the number of classes. The best result per row is in bold

| Noise Rate | PCA | BLMNN | DML | BDML (ours) |
|---|---|---|---|---|
| [0.5 0.6 0.1 0.4 0.4 0.4 0.2 0.4 0.6 0.3] | $53.68 \pm 0.43$ | $55.99 \pm 0.30$ | $67.76 \pm 0.16$ | $\mathbf{69.47 \pm 0.49}$ |
| [0.3 0.6 0.1 0.5 0.2 0.5 0.4 0.1 0.3 0.5] | $56.89 \pm 0.47$ | $59.71 \pm 0.80$ | $70.60 \pm 0.38$ | $\mathbf{72.32 \pm 0.37}$ |
| [0.1 0.3 0.6 0.4 0.5 0.4 0.5 0.1 0.1 0.2] | $60.49 \pm 0.36$ | $62.63 \pm 0.29$ | $71.43 \pm 0.50$ | $\mathbf{72.89 \pm 0.47}$ |

inference, outperforms DML, which adopt MLE for parameter estimation, across different noise levels.

### 4.1.2 CLASS-CONDITIONAL LABEL NOISE

In this experiment, we examine those methods above on $10,000$ sub-sampled data set being stratified from the original CIFAR-10 data set (Krizhevsky et al., 2012). That is, the new data subset remains the class distributions of the original training data set. The class-conditional label noise ranging from $10\%$ to $60\%$, is randomly generated by flipping the labels with corresponding proportions of data samples. We keep the previous mentioned hyper-parameters settings for each model. The results reported in Table 3 show that our proposed method BDML consistently outperforms DML and BLMNN with large margins across different noise vectors.

### 4.2 FACE RETRIEVAL ON MS-CELEB 1M WITH REALISTIC NOISY LABELS

**MS-Celeb-1M (Guo et al., 2016)** is a large scale data set (with $10M$ images of $100,000$ celebrities) that contains realistic label noise. We adopt the same setting in (Wang & Tan, 2018) to form the new training set for this experiment. In particular, we first randomly select a subset of the original training with $100K$ images of 1200 persons. This subset is then split with ratio 9:1 into a training set that contains approximately $20\%$ of label noise and a clean test set, respectively. The 4096-dimensionality feature vector of each data point in this subset is extracted by using the VG-GFace (Parkhi et al., 2015) model. We run 80 epochs, for the *BDML model* learning rate is 0.0001 with decay rate 0.5 at epoch $25, 45, 55, 65$, prior distribution for both weights and bias are Gaussian distributions with mean 0 and standard deviation 0.5; batch size is set to 256. For the *DML model*, we set learning rate to 0.00001 with decay rate 0.5 at epoch $10, 20, 40$, batch size is set to 128. For the *BLMNN model*, we follow the hyper-parameter setting suggested by authors (Wang & Tan, 2018). Table 4 shows the comparison using the mean of average precision (mAP) between our proposed BDML and other competing methods.

The results in Table 4 consistently indicate the advantage of using deep metric in real world label noisy dataset. Both DML and BDML show their better retrieval performance compared to the linear distance metric. Moreover, it is clear from Table 4 that our proposed method BDML is superior to the point estimation DML – this is evidence that the use of Bayesian inference is more robust than the point estimation when working with label noise.

## 5 CONCLUSIONS

In this paper, we proposed a novel theoretically sound Bayesian deep metric learning method to handle the issue of learning a deep model with noisy labels. To the best of our knowledge, this is the first work that introduces Bayesian inference in deep metric learning to improve the generalization capacity of two existing methods: robust Bayesian linear metric learning and point estimation deep metric learning. It is also important to note that our main goal with this paper is the introduction of an innovative framework that can improve the performance of both baselines, including BLMNN

Table 4: Experiment results for MS-Celeb-1M (mAP %). The best result per column is in bold

| Noise Rate | 0% | 20% |
|---|---|---|
| PCA | 86.21 | 82.09 |
| BLMNN | 89.16 | 85.11 |
| DML | 91.89 | 86.50 |
| BDML (ours) | **93.45** | **87.66** |

and DML. We are not aware of other SOTA methods in the scope of this paper. In the future, we plan to test our proposed method with more complicated base models (e.g., ResNet (He et al., 2016)) instead of the simple neural network to obtain better experimental results. We also plan to extend our current work to make it robust under simulated label noise (Wang & Tan, 2018) which has been shown to be more challenging than the types of label noise mentioned in our paper.

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
