# OpenReview forum: "Bayesian Metric Learning for Robust Training of Deep Models under Noisy Labels"
_ICLR.cc/2021/Conference — Reject_

### Official Review · AnonReviewer3 · 2020-10-28
**First theoretically sound approach to Bayesian deep metric learning. Improved performance for highly noisy labels compared to linear Bayesian model as well as deep point estimate model**

**Rating:** 7
**Confidence:** 2

**Review:**

# Summary

The authors present a method to employ Bayesian deep learning with a metric learning (triplet loss) objective. This is a significant extension to existing Bayesian deep learning approaches, which have focused on regression and classification approaches with vanilla L2 and cross entropy losses.

# pros:

- The proposed method appears to be first to soundly combine deep metric learning with a Bayesian approach (Bayes by Backprop) to estimate parameter uncertainties.

- In an extensive evaluation, the method is shown to deal well with highly noisy labels (up to 50%) in classification tasks like MNIST and CIFAR-10. In high noise setups, the method is shown to outperform linear Bayesian metric learning, as well as deep point estimate metric learning.

- The paper is clearly written

# cons:

- The method is not evaluated against the state of the art on the evaluated datasets, albeit the authors give a convincing reason for this, namely that the methodological novelty of the proposed approach warrants proof-of-concept results with a relatively simple neural network architecture

# comments:

- It would be interesting to hear in more detail what motivated the authors' choice of Blundell's approach to uncertainty estimation as compared to e.g. the Dropout- / Deep Gaussian Process-based works by Yarin Gal et al.

- The abstract mentions "simulated label noise" as something the authors have considered, whereas the experiments mention "synthetic" and "realistic label noise", and conclusion refers to "simulated label noise" in the context of future work. Could the authors please clarify, and add a more thorough description of the respective types of noise?

- I wasn't able to fully follow the theoretical derivation of the authors on how the approach of Blundell et al. (2015), proposed for vanilla regression problems, transfers to a metric learning setup. Hence I have low confidence in my review.

---

> ### Author Response · Authors · 2020-11-17
> **Many thanks for the positive evaluations for our paper**
>
> R3: The method is not evaluated against the state of the art on the evaluated datasets, albeit the authors give a convincing reason for this, namely that the methodological novelty of the proposed approach warrants proof-of-concept results with a relatively simple neural network architecture
>
> Response: our main goal with this paper is the introduction of an innovative robust framework that can improve the performance of Bayesian linear metric learning and point estimate deep metric learning w.r.t several tasks including image classification & retrieval. In the scope of this paper, we are not aware of other SOTA methods in the field that work for these data sets. We’ll leave that for the future work of the paper.
>
> R3: (Motivation of the choice of Blundell's approach to uncertainty estimation as compared to e.g. the Dropout- / Deep Gaussian Process-based works by Yarin Gal et al.)
>
> Response: Although those approaches can be used to approximate the posterior of the network parameter, as mentioned in the paper, we decided to employ the Blundell’s approach since it allows us to sample from a Bayesian neural network using the typical backpropagation technique.
>
> R3: (The abstract mentions "simulated label noise" as something the authors have considered, whereas the experiments mention "synthetic" and "realistic label noise", and the conclusion refers to "simulated label noise" in the context of future work. Could the authors please clarify, and add a more thorough description of the respective types of noise?)
>
> Response: Thank you for pointing that out. We actually mean  "synthetic" and "realistic label noise" in the abstract, and "simulated label noise" is for our future work. We will fix that in the modified version of the paper.
>
> R3: (I wasn't able to fully follow the theoretical derivation of the authors on how the approach of Blundell et al. (2015), proposed for vanilla regression problems, transfers to a metric learning setup. Hence I have low confidence in my review.)
>
> Response: We believe that all of the theoretical results in our paper were mathematically justified.

---

### Official Review · AnonReviewer1 · 2020-10-29
**Reviewer1 Comments**

**Rating:** 3
**Confidence:** 4

**Review:**

Briefing:

This paper proposes a new Bayesian metric learning method, robust to noisy labels. The paper introduces a variational formulation for incorporating the Bayesian framework to triplet-loss training, with supporting experiments.

Strong points

The paper proposes fancy variational derivation for the Bayesian frameworks for triplet-loss training.

Weak points

(1) An essential reference [1] and other triplet ablations [2] seems to be missing.

[1] Lin, Xudong, et al. "Deep variational metric learning." Proceedings of the European Conference on Computer Vision (ECCV). 2018.

[2] Duan, Yueqi, et al. "Deep adversarial metric learning." Proceedings of the IEEE Conference on Computer Vision and Pattern Recognition. 2018.


(2) Connected to (1), Experiments do not seem to be enough for supporting the superiority of the proposed method.


Comments:

(1) The author should clarify the difference between [1]. [1] also uses variational formulation for the triplet loss.

(2) Experimental setup seems to be not enough. Along with [1], typical retrieval dataset such as CUB-200- 2011, Stanford Online Products dataset, Cars196 dataset should be concerned.

(3) Along with [1] and [2], the author should compare recent triplet-loss-based embeddings with the mentioned typical retrieval datasets.

Note:

The paper seems to miss the related works that must compare. The paper also does not seem to include typical retrieval tasks to compare with other metric learning methods, which has been standard experiments in this field. Unless a clear explanation for this issue are provided, the reviewer cannot avoid rejecting this paper.

---

> ### Author Response · Authors · 2020-11-17
> **Respond to R1**
>
> R1:  (Differences between [1] , [2] and our paper)
>
> Response: Although [1] also employ the variational formulation for the triplet loss, the key difference between our proposed method and the one in [1] can be described as follows:
>
> i) In [1], the variational Inference is adopted for estimating the distribution of latent variable not for the model parameters
>
> ii) [1] does not focus on the robustness of the model against noisy label dataset, instead, they try to build a model that is robust against input noise with high intra-class variance without theoretical guarantee.
>
> iii) [1] focuses on the retrieval task only; while our proposed method is empirically verified to work for both classification & retrieval on several data sets containing different types of label noise.
>
> Regarding [2] that aims to minimize the "generalization bound for DML with Dropout" by using a new regularization term that explicitly improves the generalization performance. Similar to [1], [2] does not consider the robustness under noisy labels, either. Moreover, [2] does not actually use variational inference -- instead, [2] employs deep adversarial training in its deep metric learning framework.
>
> We will add that to the revised version of our paper with the clarification above.
>
> R1: (Experimental on typical retrieval datasets)
>
> Response: This probably is a misunderstanding about the main goal of our paper.
> As mentioned in the paper, our main goal with this paper is the introduction of an innovative framework that can improve the performance of Bayesian linear metric learning and point estimate deep metric learning w.r.t several tasks including image classification & retrieval. We are not aware of other methods in the field that work for these retrieval data sets.
>
> [1] Lin, Xudong, et al. "Deep variational metric learning." Proceedings of the European Conference on Computer Vision (ECCV). 2018.
>
> [2] Duan, Yueqi, et al. "Deep adversarial metric learning." Proceedings of the IEEE Conference on Computer Vision and Pattern Recognition. 2018.

---

### Official Review · AnonReviewer4 · 2020-10-30
**I can't find any contributions that can conquer me, thus, this paper should not be accepted by ICLR2021**

**Rating:** 4
**Confidence:** 5

**Review:**

This paper introduces a Bayesian deep metric learning framework that is robust against noise labels. The proposed method is inspired by the BLMNN (Wang & Tan, 2018), deep metric learning (Hoffer & Ailon, 2015; Hu et al., 2015; Wang et al., 2017; Lu et al., 2017; Do et al., 2019), and Bayes by Backprop (Blundell et al., 2015).  Different from BLMNN that only considers a linear metric learning, the authors’ framework can handle non-linear deep metric learning, which is useful for many real-life applications. Moreover, directly applying the variational Bayes learning (Wang & Tan, 2018) in deep learning is challenging since it requires sampling from a distribution of the neural network parameters. Instead, The author adapt the variational inference by Blundell et al. (Blundell et al., 2015), which allows to efficiently sample the parameters of a Bayes neural networks by using a backpropagation-compatible algorithm. They also theoretically show the robustness of the proposed method when working with label noise. The experimental results on several noisy data sets show that their novel proposed method can generalize better compared to the linear BLMNN (Wang & Tan, 2018) and the point estimation-based deep metric learning (Hoffer & Ailon, 2015; Lu et al., 2017), especially when the noise level increases. In my opinion, the main novelty of this

+ves:
+ The idea of using the variational inference by Blundell et al. (Blundell et al., 2015) to derive Bayesian version of deep metric learning is interesting.

+ Overall, the paper is well-written and well-organized. In particular, the Related Work section has a nice flow and puts the proposed method into context. Despite the method having limited novelty (sliding window instead of a growing window), the method has been well motivated in Sections 1 and 3.

+ The theoretical analysis section is completed. And the results section is well structured. It's nice to see the advantages of the proposed method over other compared methods under different datasets with label noise

Concerns:
- The key concern about the paper is the contributions of this paper. In my opinion, the novelty of this paper is limited. The main contribution is to propose a Bayesian version of deep metric learning. In fact, the main strategies and motivation from (Blundell et al., 2015) and (Wang & Tan, 2018). I cannot find any original contributions such as sampling methods and novel loss function in this paper. In a word, this paper only combines variational inference by Blundell et al. (Blundell et al., 2015) with deep metric learning without any original techniques.

- Lack of sufficient analysis to show that your method is robust to label noise. In modeling, the authors did not consider any label information, but the model is robust to label noise. This is questionable. In addition, the result of Theorem 1 is meaningless.
I don’t want to waste my time explaining why your theoretical results are meaningless, but I recommend some papers as follows:
[1] Deep Metric Learning: The Generalization Analysis and an Adaptive Algorithm
[2] Robustness and Generalization for Metric Learning

-According to my experience, the Bayesian version of a model should be much higher than the original model. But the author's experimental results are not satisfied.

I can't find any contribution that can conquer me, thus, this paper should not be accepted by ICLR2021

---

> ### Author Response · Authors · 2020-11-17
> **Respond to R4**
>
> R4: (Novelty and the contributions of this paper)
>
> Response: Please see our response to R2’s comment about the novelty of our paper. We would like to emphasize that the main goal of our paper is to introduce the use of Bayesian inference framework aiming a novel robust deep metric learning against noisy labels. As mentioned in the paper, our main contributions can be summarized as follows:
>
>  Represent a triplet-based deep metric learning using Bayesian inference to introduce more robust feature spaces for several learning tasks (including classification, retrieval).
>
> Employ the variational inference framework (Blundell et al., 2015) to efficiently sample from a Bayesian neural network using backpropagation technique.
>
> Provide theoretical as well as empirical guarantees for the robustness training under label noise of the proposed method.
>
> R4: (label information was not considered)
>
> Response: In our model, the label information is actually used to form the triplet, to define a noisy label triplet, and then triplet loss (Sec. 3.1). In particular, a triplet consists of an anchor, a positive sample (that has the same label), and a negative sample (that has a different label).
>
> R4: (Lack of sufficient analysis to show that your method is robust to label noise)
>
> Response: The theoretical results in Theorem 1 indicate that our proposed method is robust against label noise given a particular noise level (presented by the number of noisy label triplets in the problem setup) in the sense of the Bayesian inference framework. Without those results, we can not guarantee that the learning performance won’t be significantly affected, even with a small noise level -- this is crucial due to the sensitivity of deep models under noisy labels.
>
> R4: (Experimental results are not satisfying)
>
> Response: The experimental results in our paper consistently show that our proposed method provides better classification and retrieval performances with respect to several types of noisy labels compared to the baselines, especially when the noise level increases. We believe that those results convincingly support  the main claims of our paper,
>
> [1] Robustness and Generalization for Metric Learning
>
> [2] Deep Metric Learning: The Generalization Analysis and an Adaptive Algorithm.

---

### Official Review · AnonReviewer2 · 2020-11-05
**This paper should not be accepted by ICLR2021**

**Rating:** 5
**Confidence:** 4

**Review:**

This paper proposes a robust Bayesian deep metric learning framework against noise label inspired the BLMNN (Wang & Tan, 2018), deep metric learning (Hoffer & Ailon, 2015; Hu et al., 2015; Wang et al., 2017; Lu et al., 2017; Do et al., 2019), and Bayes by Backprop (Blundell et al., 2015). Directly applying the variational Bayes learning (Wang & Tan, 2018) in deep learning is challenging since it requires sampling from a distribution of the neural network parameters. Instead, this paper adapts the variational inference by Blundell et al. (Blundell et al., 2015), which allows to efficiently sample the parameters of a Bayes neural networks by using a backpropagation-compatible algorithm. The experimental results on several noisy data sets show that our novel proposed method can generalize better compared to the linear BLMNN (Wang & Tan, 2018) and the point estimation-based deep metric learning (Hoffer & Ailon,
2015; Lu et al., 2017), especially when the noise level increases.

Pros:
1.  This paper is well-organized and well-written.
2.  Adapting the variational inference by Blundell et al. (Blundell et al., 2015) for Bayesian DML sounds good.
3.  The theoretical analysis is completed (though meaningless)

Cons:
1.	The novelty is limited (just using sliding window to instead growing window). It is more like a combination of variational inference (Blundell et al., 2015) and DML.
2.	The results of Theorem 1 are meaningless. Some papers [1] Robustness and Generalization for Metric Learning. and [2] Deep Metric Learning: The Generalization Analysis and an Adaptive Algorithm. may help you to understand this point.

---

> ### Author Response · Authors · 2020-11-17
> **Respond to R2**
>
> R2: (Novelty of the paper)
>
> Response: Our approach is based on an extension of the triplet loss function to a Bayesian inference framework to robustly train deep models with several types of label noise. Wang & Tan only focus on a simple linear Mahalanobis distance that can not capture the nonlinear relationship of data points in deep metric learning. Moreover, directly applying the idea of Wang & Tan for deep metric learning is challenging due to the complexity of the sampling task in the parameter space. We, therefore, adapt the Bayesian linear metric learning to propose the Bayesian deep metric learning that is robust against noisy labels. The variational inference (Blundell et al. 2015) is employed in our framework targeting an effective parameter sampling procedure. To the best of our knowledge, this is the first theoretically sound Bayesian deep metric learning that is robust against noisy labels.
>
> R2: (Role of the results of Theorem 1)
>
> Response: The theoretical results in Theorem 1 indicate that our proposed method is robust against label noise given a particular noise level (presented by the number of noisy label triplets in the problem setup) in the sense of the Bayesian inference framework. Without those results, we can not guarantee that the learning performance won’t be significantly affected, even with a small noise level -- this is crucial due to the sensitivity of deep models under noisy labels.
>
> R2: (Robustness in [1] and [2])
>
> Response: First, we would like to emphasize that the main idea of our paper is to introduce the use of Bayesian inference targeting a robust deep metric learning under noisy labels, while both [1] & [2] do not take the robustness against noisy labels into account.
>
> We agree with the reviewer that the definition of the robustness of metric learning in those papers is interesting since they can provide an explicit generalization bound. However, they are not associated with the robustness under noisy labels considered in our paper. We, therefore,  plan to work on a more concrete definition/verification robustness based on those papers, e.g., to provide an upper bound for the generalization error when training with noisy labels.
> We will add that to the revised version of our paper with the clarification above.
>
> [1] Robustness and Generalization for Metric Learning
>
> [2] Deep Metric Learning: The Generalization Analysis and an Adaptive Algorithm.

---

### Decision · Program_Chairs · 2021-01-07
**Final Decision**

**Decision:**

Reject

**Comment:**

Reviewers have commented on the lack of novelty of the paper as it reads only as applying the variational inference framework of Blundell et al. (2015) to deep metric learning (R2 and R4). Furthermore, the paper has not properly positioned itself when compared to previous works on "Deep variational metric learning" and "Deep adversarial metric learning" (R1) and other previous literature that have studied robustness for metric learning. The argument on robustness to noisy labels needs to be expanded and better fleshed out in a future version of the paper.